# Comorbidities and Laryngeal Cancer in Patients with Obstructive Sleep Apnea: A Review

**DOI:** 10.3390/medicina59111959

**Published:** 2023-11-06

**Authors:** Beata Kiss, Cristian Mircea Neagos, Gabriela Jimborean, Hédi Katalin Sárközi, Mioara Szathmary, Adriana Neagos

**Affiliations:** 1Department of Otorinolaryngology, Emergency County Hospital Targu Mures, George Emil Palade University of Medicine, Pharmacy, Science and Technology, 540136 Târgu Mures, Romaniacristian.neagos@gmail.com (C.M.N.); 2Pneumologic Department, George Emil Palade University of Medicine Science and Technology, 540136 Târgu Mures, Romania; gabriela.jimborean275@gmail.com (G.J.); baloghedikatalin@yahoo.com (H.K.S.); mioara.szathmary@gmail.com (M.S.)

**Keywords:** obstructive sleep apnea, head and neck cancer, hypermetabolic syndrome, cardiovascular diseases, stroke

## Abstract

*Introductions*: The global prevalence of obstructive sleep apnea shows that this disease appears in 1 billion people, with the prevalence exceeding 50% in some countries. Treatment is necessary to minimize negative health impacts. Obstructive sleep apnea (OSA) is defined as a cause of daytime sleepiness, as well as a clinical manifestation of sleep-disordered breathing. In the literature, there are numerous controversial studies regarding the etiology of this condition, but it is universally accepted that reduced activity in the upper airway muscles plays a significant role in its onset. Additionally, OSA has been associated with a series of comorbidities, such as type II diabetes, metabolic syndrome, and cardiovascular and pulmonary conditions, as well as head and neck tumors, especially oropharyngeal and laryngeal tumors. This is a review of the subject of OSA that considers several aspects: an analysis of the comorbidities associated with OSA, the involvement of tumor pathology in the onset of OSA, and the association of OSA with various types of laryngeal cancers. Additionally, it includes an evaluation of postoperative and medical outcomes for patients with OSA and laryngeal tumors treated surgically and medically, including chemotherapy. *Relevant Sections*: By taking into consideration the stated objective, a systematic analysis of the available literature was conducted, encompassing the PubMed, Medline, and Scopus databases. The evaluation was based on several keywords, including head and neck cancer, diabetes, diabetic, overlap syndrome, cardiovascular conditions, laryngeal neoplasm, radiotherapy, and chemotherapy, as well as the concept of quality of life in laryngectomized patients and patients with OSA. *Discussions*: The review evaluates the involvement of OSA in the presence of comorbidities, as well as the increased incidence of OSA in patients with laryngeal cancer. It is important to note that surgical and post-surgical treatment can play a significant role in triggering OSA in these patients. *Conclusions*: The studies regarding the correlations between OSA, comorbidities, and head and neck tumors indicate a significantly increased risk of OSA in association with conditions such as diabetes, metabolic syndrome, cardiovascular diseases, and head and neck tumors, particularly laryngeal tumors. This association has a physio-pathological basis. The various surgical methods followed by radiation and chemotherapy for tumor treatment do not exclude an increased risk of developing OSA after treatment. This significantly influences the quality of life of patients who survive these types of tumors. *Future directions*: Due to the multiple comorbidities associated with OSA, the extension of polysomnography associated with investigations during sleep, such as drug-induced sleep endoscopy, represents a tendency for the early diagnosis of this pathology, which affects the quality of life of these patients. Patients with head and neck cancer are at high risk of developing obstructive sleep apnea; this is why it is necessary to expand the polysomnographic investigation of these patients after surgical procedures or after radiotherapy and chemotherapy.

## 1. Introduction

Obstructive sleep apnea (OSA) is a chronic condition that occurs due to the collapse of the upper airway, resulting in hypoxemia and intermittent awakening, known as arousal. The prevalence of this condition is 3–7% in men and 2–5% in women. The risk of developing OSA increases with age, with risk rising to 30% in men over the age of 80, which may be caused by a reduction in deep sleep.

The global prevalence of obstructive sleep apnea shows that this disease appears in 1 billion people, with the prevalence exceeding 50% in some countries. Treatment is necessary to minimize the negative health impact. This condition represents a major risk for triggering diseases such as hypertension, ischemic heart disease, diabetes, arrhythmias, or mental illnesses. As a result, it affects the quality of life in a direct way [1].

When taking into account the global magnitude of the aforementioned pathology, the diagnosis and treatment of sleep pathology represents a large economic burden. In countries with well-developed healthcare systems, efficient treatment aims to decrease the need for office visits, whereas, for low- and medium-income countries, where resources for diagnosis and treatment are scarce, the emphasis needs to be on providing cost-effective care.

When analyzing patients with OSA, we underline the importance of the onset of symptoms. When we talk about a patient with OSA, we are referring to the way this condition appears and the possibility of diagnosis through polysomnography, as well as the treatment modalities.

Investigation methods and local examination, such as nasopharyngoscopy, oropharyngoscopy, and laryngoscopy, allow for the evaluation of the upper airway, the oral cavity, the pharynx, the larynx, the salivary glands, and the thyroid region [1].

Tumors located at these levels or involving the para- and retropharyngeal spaces cause a narrowing of the respiratory space, sometimes up to its complete obstruction, with the onset of OSA. Tumor formations can be diagnosed during a clinical examination, but often, they cannot be visualized because they infiltrate the deeper tissues, and the overlying mucosa remains unmodified [1].

If we are talking about the role of the larynx in the occurrence of OSA, we need to understand the normal function of the larynx during sleep, as well as the contribution of the larynx to the onset of OSA as a possible level of airway obstruction [2]. It is important to understand the normal physiology of the larynx and the contribution of laryngeal dysfunction to the appearance of clinical syndromes, which can be evaluated through fiberoptic examination methods. This fiberoptic examination can be used to assess the airway, even down to the subglottic level, and it can be performed while the patient is awake or during sleep. This method is used to identify structures within the airway and to evaluate its anatomy, as well as any existing changes in patients with OSA. 

The larynx and neighboring structures play a role in the pathophysiology of OSA in adults, with a recognized multi-level contribution to this condition. Those patients with head and neck cancer have a higher incidence of OSA compared with the general population. The frequency is 59.78%. When discussing the association between the larynx, laryngeal pathology, and OSA, this leads to the conclusion of understanding their involvement in the pathophysiological mechanisms of OSA. Regarding the larynx’s involvement in the onset of OSA, the literature is limited but still demonstrates a relationship between treatment strategies for patients with OSA who also have laryngeal pathology [1,2,3].

OSA is a chronic condition associated with increased mortality and morbidity due to cardiovascular, metabolic, and neurocognitive disorders. It is linked to repetitive episodes of hypoxemia, where intermittent drops in oxygen levels occur. Intermittent hypoxemia triggers the activation of angiotensin II receptors, leading to increased aldosterone levels, elevated levels of endothelin, and vasoconstriction. Additionally, oxidative stress plays a role in this process. Sleep fragmentation and the occurrence of micro-arousals contribute to the cardiovascular pathophysiology in OSA, with hypoxemia being a factor in cardiac and vascular dysfunction [4]. The patients with obstructive sleep apnea appear to have predominantly anatomic and physiologic causes, and patients with a narrow transpalatal airway appear to have a collapse or an obstruction in the upper airway. We should always analyze in terms of the degree of obstruction and the anatomical modifications in relationship with the polysomnographic investigations.

Craniofacial abnormalities are considered a primary factor in the development of OSA and must be identified alongside other factors, including the body mass index (BMI), Mallampati index, or retrognathia. These factors, in addition to laryngeal pathology, contribute to the onset of OSA, and their evaluation through flexible fiberoptic examination should be considered. Craniofacial abnormalities, as a primary factor in the development of OSA, must be identified. Other predisposing causes are represented by sex, age, race, and obesity. Men have a higher risk of developing OSA than women, although this difference disappears once women reach menopause [5]. There is also a correlation between obesity (body mass index > 30 kg/m^2^) and OSA. A 10% increase in body mass increases the apnea-hypopnea index (AHI) by 32%, resulting in a six-fold increase in moderate to severe OSA.

Other possible causes for the onset of OSA are the obstruction or narrowing of the respiratory tract caused by nasal abnormalities, such as deviated nasal septum or turbinate hypertrophy, palatine tonsil hypertrophy, macroglossia, jaw misalignment, enlarged or elongated uvula, and benign or malignant tumors of the respiratory tract [5,6].

A physical examination includes the measurement of the neck circumference (greater than 17 inches for men or 16 inches for women), determining BMI, Mallampati scores, and a complete otorhinolaryngological examination [5]. Investigation methods and local examination, such as nasopharyngoscopy, oropharyngoscopy, and laryngoscopy, allow for the evaluation of the nasal passage, the oral cavity, the pharynx, and the larynx [1]. Physical examination and endoscopy are followed by polysomnography, which is currently the gold standard for diagnosing OSA.

The larynx and neighboring structures play a role in the pathophysiology of OSA in adults, with a recognized multi-level contribution to this condition. Regarding the larynx’s involvement in the onset of OSA, the literature is limited but still demonstrates a relationship between OSA and laryngeal pathology [1,3]. If we are talking about the role of the larynx in the occurrence of OSA, we need to understand the normal function of the larynx during sleep, as well as the contribution of the larynx to the onset of OSA as a possible level of airway obstruction [2]. It is important to understand the normal physiology of the larynx and the contribution of laryngeal dysfunction to the appearance of clinical syndromes, which can be evaluated through fiberoptic examination methods. This fiberoptic examination can be used to assess the airway down to the subglottic level, and it can be performed while the patient is awake or during sleep. This method is used to identify strictures within the airway and to evaluate its anatomy, as well as any existing changes in patients with OSA [5,7].

Due to the fact that head and neck neoplasms, and particularly laryngeal neoplasms, are less commonly associated with OSA, there is a concern that the assessment of patients with neoplasms may be lacking [1,2]. According to the American Cancer Society, the lifetime risk of developing laryngeal cancer for men is 1 in 190, and for women, 1 in 830. A total of 12,380 new cases of laryngeal cancer are estimated to occur in the United States of America in 2023, although the rate of new cases is falling worldwide.

This study represents a review of the subject of OSA, considering several aspects, as follows: an analysis of the comorbidities associated with OSA, the involvement of tumor pathology in the onset of OSA, and the association of OSA with various types of laryngeal cancers. Additionally, it includes an evaluation of postoperative and medical outcomes for patients with OSA and laryngeal tumors treated surgically and medically, including chemotherapy. 

## 2. Relevant Sections

This review is structured around a collection of publications systematically organized into four chapters: data selection related to the association between OSA and craniofacial modifications, the evaluation of frequently encountered comorbidities, the correlation regarding laryngeal and head and neck tumor pathology with OSA, and the correlation of therapeutic outcomes in patients with OSA with head and neck and laryngeal tumors. The data from the literature were collected through a systematic analysis of the results obtained from clinical studies.

This work analyzes the results in the field of sleep pathology, encompassing several subdomains, such as specific evaluation methods for patients with sleep apnea syndrome, in order to determine the site of obstruction, sleep apnea syndrome and comorbidities, and sleep apnea and head and neck tumors, with an evaluation of surgical and nonsurgical therapeutic outcomes, as well as the study of sleep pathology in laryngectomized patients. The review also analyzes the existing link between sleep apnea syndrome and comorbidities, such as laryngeal cancer, and also discusses the diagnostic methods used. It also explores how both surgical and nonsurgical treatments, like radiotherapy and chemotherapy, influence sleep apnea syndrome. A literature search was performed using international databases like PubMed, Scopus, Index Copernicus, and the Web of Science. We have used terms like obstructive sleep apnea, comorbidities, cardiovascular diseases, laryngeal cancer and OSA, and head and neck cancer and OSA. 

The relevant sections of this review comprise the following parts: the importance of flexible nasopharyngoscopy for the evaluation of the upper airway and the involvement of anatomical structures in obstructive sleep apnea, complete with a section about obstructive sleep apnea and comorbidities, where we have analyzed what the most common comorbidities are and how they are associated with sleep pathology, especially with obstructive sleep apnea. The evaluation was based on several keywords, including “head and neck cancer”, “diabetes”, “diabetic”, “overlap syndrome”, “cardiovascular conditions”, “laryngeal neoplasm”, “radiotherapy”, “chemotherapy”, and “laryngeal cancer”, as well as “quality of life” in laryngectomized patients and patients with OSA. Only publications written in the English language were included in the present review. The searches were conducted in September 2022. All studies referring to OSA in combination with the aforementioned terms were included in the review. A total of 30 studies were included in the present review. The review process was conducted with the participation of all authors. The extracted data include patient demographics, cancer details, comorbidities, and OSA and laryngeal cancer treatment and outcomes. The present study is a review aiming to provide an overview of the topic of OSA in relation to comorbidities, especially laryngeal cancer, and a meta-analysis was not conducted. Various pathologies, such as diabetes, ischemic heart disease, heart disorders, and lung pathology, were extensively analyzed, showing the connection between them and obstructive sleep apnea. Another relevant section of this review is the one related to neck cancer, and especially laryngeal cancer, referring both to the involvement of the laryngeal structure in maintaining the upper airway and to the way in which laryngeal cancer is involved in the occurrence of obstructive sleep apnea. 

The concept of the management of obstructive sleep apnea in patients with head and neck cancer analyzes how the partial surgical therapeutic techniques, as well as radio-chemotherapy treatment, act on obstructive sleep apnea.

## 3. Discussions

In order to collect physiologic parameters during sleep, it is necessary to perform polysomnography. A polysomnogram is a procedure that offers information about sleep quality, airflow, and respiratory effort. It is considered to be the gold standard in terms of evaluating sleep disorders and the degree of the apnea-hypopnea index. The value of the apnea–hypopnea index in terms of obstructive sleep apnea syndrome is classified into mild, moderate, and severe sleep apnea. 

### 3.1. Flexible Nasopharyngoscopy in Assessing the Upper Airway, and the Involvement of Anatomical Structures in Obstructive Sleep Apnea Syndrome

In order to evaluate the upper airway and its changes, flexible nasopharyngoscopy is used, mostly because OSA is primarily caused by a combination of factors that result in the narrowing of the upper airway. In a small number of patients, this is caused by treatable but undiagnosed tumors [1]. In 79 reported and published cases, lipomatous tumors of the parapharyngeal and retropharyngeal spaces lead to airway obstruction [1]. Evaluation using nasopharyngoscopy is recommended in the literature, with research indicating that isolated transoral examination allows for the proper visualization of the region in only 53.2% of cases [1]. Studies may be limited due to inconsistent reports, as there are discrepancies between the results of polysomnographic investigations and fiberoptic examination. The sites of anatomical modifications identified through this method are as follows: the parapharyngeal region: 44.3%, oropharynx: 22.6%, larynx: 11.4%, nasopharynx: 6.3%, deep cervical space: 5%, oral cavity: 3.8%, and rhinosinus cavities: 2.5% [1]. In patients with upper airway neoplasms, the evaluation of the airway through pharyngoscopy can help demonstrate the clinical correlation between these pathologies [1].

When speaking about the cervical region, especially the involvement of the larynx in the onset of OSA, and when analyzing the specialized literature consisting of 17 articles, the following observations were made: intraluminal negative pressure and the anatomy of the upper airway contribute to its dynamic closure at the end of expiration during sleep [3]. The presence of active forces of dilator muscles can counteract this process [3], with innervation of the genioglossus muscle provided by the hypoglossal nerve being a fundamental element. Experiments on animals demonstrate the contribution of mechanoreceptors in activating the genioglossus muscle [2]. Examinations of the posterior part of the larynx reveal a decrease in abduction during sleep, and polysomnography shows a high-pressure gradient between the supraglottic and esophageal probes, indicative of upper-airway obstruction. The literature states that the collapsing pressure is 6.4 cm of water (cm H_2_O) [2,3]. The involvement of laryngeal structures in OSA can lead to the development of obstructive sleep apnea in patients with laryngeal cancer, systemic degenerative atrophy, and neurological dysfunction. It can be asserted that sensory dysfunctions contribute to the occurrence of sleep apnea, and the irradiation of the mucosa, causing the alteration of chemoreceptors, negatively affects dilator muscles [3]. Regarding sleep endoscopy as a method for evaluating the upper airway, it has been demonstrated that the normal phase of vocal cord abduction during inspiration followed by adduction during expiration occurs similarly during sleep in morphologically normal patients [2]. There is a bidirectional relationship between obstructive sleep apnea syndrome and the larynx, which can be a possible site of obstruction. Various neurological conditions must also be considered in the evaluation of adults with obstructive sleep apnea. Sensory laryngeal disorders can exacerbate OSA symptoms, and the evaluation of upper airway collapse at the laryngeal level must be assessed through pharyngoscopy and endoscopy during sleep. Multi-level obstruction is described in 72% of the cases evaluated in the literature, and there may be an association with elevated apnea-hypopnea index (AHI) values in these patients [2]. The intermittent hypoxemia observed in patients with OSA is demonstrated by multiple studies, summarizing the presence of neuronal damage that can be mitigated through CPAP (continuous positive airway pressure) therapy. In the case of tumors, intermittent hypoxemia has a proangiogenic effect and promotes tumor growth, as demonstrated in in vitro studies [4]. In this context, when considering the SpO2 < 90% hypoxemia index as a marker of OSA severity, it has been shown that cancer patients with OSA are eight times more likely to experience mortality than those without sleep apnea, taking into account factors such as obesity, smoking, and other collateral factors [4]. When referring to anatomical and functional parameters, such as facial morphology, oral cavity, oropharynx, and dentomaxillary malformations, statistical analysis using the Spearman correlation indicates that in the majority of patients, obstructive sleep apnea appears in male patients [5] with a body mass index 29 ± 5 kg/m^2^ [5]. Cephalometric measurements from the literature show variability in the values of MPH (mandible plane-hioid distance) at 29.14%, posterior airway space at 30.04%, and uvula at 10.47%, as evaluated through pharyngoscopy [6].

### 3.2. Obstructive Sleep Apnea and Comorbidities

Obstructive sleep apnea is frequently associated with comorbidities, such as metabolic syndrome, diabetes mellitus, and pulmonary-, cardiac-, and neuropsychiatric conditions. It is widely accepted that OSA is an independent risk factor for most of these comorbidities, but recent studies demonstrate that some of these comorbidities act as predisposing factors in triggering OSA. A bidirectional relationship is especially demonstrated for cardiac conditions, metabolic syndrome, and stroke. This is explained by fluid retention and its distribution in the brain, as well as the neural mechanisms involved in the onset of diabetes and stroke [7].

Obesity, obstructive sleep apnea, and gastroesophageal reflux contribute to the worsening of bronchial asthma symptoms. In 50% of overweight and obese patients, bronchial asthma is associated with these conditions, with obesity playing a role in triggering bronchial asthma. Weight loss is a viable solution in the treatment of asthma in obese individuals, in combination with bariatric surgery and medication use, which also improves quality of life. Most studies demonstrate that weight reduction influences asthma outcomes. Physical exercise is key to reducing body weight in asthmatics, impacting cardiovascular conditions, type 2 diabetes, cancer, and depression equally.

There is a connection between asthma and sleep apnea, with the prevalence of OSA in patients with asthma being higher compared to those without asthma. The prevalence of OSA in asthmatic patients has been demonstrated to be 49.5% [8]. The relationship between OSA and the severity of asthma is less clearly understood. Studies show an increased risk of OSA in subjects with severe bronchial asthma compared to those with moderate asthma. CPAP therapy in these patients has shown a reduction in daytime symptoms after obesity adjustment, influencing the quality of life and demonstrating the impact of this therapy on the quality of life in asthmatics who use CPAP [8,9].

Chronic obstructive pulmonary disease (COPD) and obstructive sleep apnea (OSA) are two respiratory disorders, and cardiovascular disease is the primary comorbidity present in both conditions. Patients with COPD experience reduced sleep quality due to associated ventilation disorders and impaired gas exchange during sleep. The coexistence of OSA with COPD is referred to as an overlap syndrome [9]. Patients with overlap syndrome exhibit high levels of hypoxemia and hypercapnia, leading to the development of pulmonary hypertension, right ventricular dysfunction, and increased morbidity [9]. In overlap syndrome, the literature describes a relationship between pulmonary hyperventilation and sleep quality, which can be assessed through pulmonary function studies [9,10]. An evaluation of patients with overlap syndrome revealed an association between sleep efficiency and the apnea-hypopnea index (AHI) and, consequently, between sleep efficiency and pulmonary hyperventilation [10]. The mechanism remains unclear, but it is considered necessary to establish a treatment aimed at reducing pulmonary hyperventilation to improve sleep efficiency. In patients with chronic lung diseases, sleep apnea, and overlap syndrome, systemic inflammatory phenomena occur within cardiovascular conditions. This involves the development of endothelial dysfunction, leading to the release of inflammatory cytokines, such as necrotizing tumor factors. Monocytes and T cells infiltrate the vascular wall and transform into macrophages. Both OSA and COPD are associated with increased activation of inflammatory cells and the mechanisms associated with atherosclerosis [10]. There are many interrelated phenomena between hypoxia and hypercapnia in the inflammatory response. Hypoxia contributes to atherosclerosis by stimulating angiotensin.

In obstructive sleep apnea (OSA), intermittent hypoxemia, sleep fragmentation, and increased sympathetic nervous system activity are common occurrences. Systemic inflammatory phenomena develop as a broad response to intermittent hypoxia, which is specific to this condition [9,10,11,12]. As a result of all the analyses conducted, overlap syndrome is believed to be associated with significant cardiovascular complications due to hypoxemia and hypercapnia. Early recognition, diagnosis, and treatment with continuous positive airway pressure (CPAP) therapy is shown to significantly reduce morbidity and mortality in these conditions [9]. There is a strong connection between obesity, obstructive sleep apnea syndrome, gastroesophageal reflux, and pulmonary pathology, especially with asthma. As such, bronchial asthma has a prevalence of 16.8% in obesity, 35.1% in obstructive sleep apnea (OSA), and 4.65% in gastroesophageal reflux disease. The prevalence of OSA is 49.5% in individuals with asthma, 21.8% in obese women, and 65% in those with gastroesophageal reflux [7,9]. Obese patients often exhibit reduced respiratory volume, particularly in the supine position. Intermittent hypoxemia increases insulin resistance and leptin levels, both of which can be improved by weight loss, leading to a reduction in the apnea-hypopnea index (AHI) and OSA severity. CPAP therapy, in conjunction with calorie restrictions, is associated with favorable outcomes in these cases [7,9,10].

OSA represents a chronic condition and a frequent comorbidity in patients with type 2 diabetes mellitus [2]. Intermittent hypoxemia and sleep fragmentation lead to changes in glucose metabolism and the development of type 2 diabetes [2]. This association with diabetes is supported by therapeutic outcomes achieved with CPAP (continuous positive airway pressure) therapy in OSA patients. Studies using new definition criteria have shown a prevalence of 23% in women and 49% in men for moderate to severe OSA. When using the new scoring criteria, a strong association between OSA and comorbidities such as type 2 diabetes, metabolic syndrome, hypertension, depression, and cardiovascular diseases is confirmed [11].

The literature supports the fact that type 2 diabetes accounts for 90–95% of all cases of diabetes [11], and in the United States, 9.3% of the population has diagnosed or undiagnosed diabetes. The physio-pathological connection between diabetes and OSA is attributed to intermittent hypoxemia and sleep fragmentation, which cause metabolic dysfunctions. Experiments in this field show that sleep fragmentation and intermittent hypoxemia disrupt glucose metabolism. Exposure to intermittent hypoxemia for more than 5 h, inducing a desaturation of 24 events per hour, has been observed [11,12]. The role of sleep fragmentation on glucose metabolism has been demonstrated in experiments that monitor eye movement in REM sleep or N-REM sleep fragmentation, resulting in a 20–25% reduction in insulin sensitivity [11].

CPAP therapy is proven to be the preferred therapeutic method in cases of OSA in patients without diabetes, with prediabetes, or with type 2 diabetes. Using CPAP for a period of 1–21 weeks for more than 4 h, demonstrates the effectiveness of combining weight loss with CPAP utilization [11]. In patients with prediabetes, the use of CPAP therapy demonstrates its influence on insulin sensitivity. Studies show the role of lifestyle factors such as weight loss and physical activity combined with OSA therapy as a primary strategy for preventing type 2 diabetes [11]. For patients with type 2 diabetes, the use of CPAP for a period of 7 days shows a decrease in plasma glucose levels. The impact on glucose levels is more pronounced during the night, resulting in lower morning glucose levels. Thus, it has been proven that CPAP therapy has a similar effect to oral antidiabetic therapy [11]. In this context, future directions suggest innovative interventions for OSA treatment by maximizing CPAP therapy. This will lead to an assessment of the effect of OSA therapy on glucose metabolism [11].

OSA is also associated with cardiovascular complications such as hypertension, arrhythmia, atrial fibrillation, coronary conditions, and stroke [12]. Hypertension has a higher prevalence of 30–50% in individuals with OSA. OSA and hypertension are common conditions with multifactorial causes that often coexist [12]. When it comes to atrial fibrillation, OSA is an independent factor. Multiple retrospective studies have shown the ability of CPAP treatment to reduce atrial fibrillation. Cardiac conditions are generally associated with OSA. The literature suggests that in obese patients with cardiac disorders and reduced ejection fraction, signs of central or mixed-type apnea may appear, making the application of positive pressure necessary. Oxidative stress and systemic inflammation, contributing to coronary atherosclerosis, can lead to acute myocardial infarction due to arterial calcifications, instability, and vascular vulnerability. In these situations, CPAP therapy has proven to be effective in reducing the risk of acute myocardial infarction [12]. A 71% prevalence of stroke in patients with OSA demonstrates that OSA is not an independent risk factor for stroke. It has been established that hypercoagulability, oxidative stress, inflammation, paradoxical embolisms, and cerebral hemodynamics are associated risk factors [12].

### 3.3. Obstructive Sleep Apnea in Patients with Head and Neck Cancer

The anatomy of the upper airway can be altered by head and neck tumors, leading to the development of obstructive sleep apnea syndrome. In order to establish the correct treatment and assess these patients, it is important to evaluate these risk factors. There are studies that assess the association between tumor stages and the treatment of cancer, as well as OSA, in patients with head and neck cancer [1,13].

Head and neck cancer has a mortality rate of 20% in the United States, with risk factors including tobacco use, alcohol consumption, and viral infections such as the Epstein-Barr virus and papillomavirus [13]. The most common form of head and neck cancer is represented by squamous cell carcinoma. These tumors can lead to abnormalities in the airway and, consequently, obstructive sleep apnea syndrome [13,14]. Airway collapse with decreased oxygen saturation and micro-arousal-type awakenings associated with neurocognitive and cardiometabolic disturbances can be identified through polysomnography. The prevalence of this condition is considered to be 10–30% in adults [13].

The prevalence of obstructive sleep apnea before and after radiotherapy in patients with head and neck cancer is higher than in normal patients. The explication is the increase in retroglossal pharyngeal area after treatment. It is important to suggest that the physician who manages patients with head and neck cancer should consider the occurrence of obstructive sleep apnea before and after the treatment. 

Patients with head and neck cancer develop obstructive sleep apnea more frequently. For these patients, it is necessary to continue investigations, which are important to identify potential risk factors, along with prevention and treatment strategies.

The analysis of head and neck cancers reveals that the most common tumors include nasopharyngeal carcinomas, lymphomas, sarcomas, adenocarcinomas, and melanomas. In terms of localization, tumors were most frequently found in the oropharynx, larynx, oral cavity, and nasopharynx, as well as metastatic tumors. When it comes to tumor staging, stages 0–II correspond to an early-diagnosed condition. However, the patients participating in the studies were detected at an advanced stage (stage IV) [13].

Patients with head and neck tumors often experience multiple causes of fatigue and manifestations of hypersomnolence [14]. When it comes to studies related to the relationship between OSA and head and neck tumors, OSA has been demonstrated to exist, especially post-surgery. Additionally, OSA has been identified in patients who have undergone partial supracricoid laryngectomy compared to vertical laryngectomy, and there is an increased prevalence of OSA in relation to surgery compared to nonsurgery [13]. Most studies (85%) use a personal assessment of sleep variables, with only a few using quality-of-life assessment scales. Some studies suggest that the Epworth sleepiness scale (ESS) can be used alone or in combination with other scales [13]. A correlation between ESS and the apnea-hypopnea index (AHI) has been found, and three studies evaluating ESS in head and neck cancer patients show a statistically significant degree of correlation with OSA, although this is observed in only one study [13]. Various approved forms have been used to assess the quality of life of patients with OSA. The association between OSA and head and neck cancers has an understandable explanation, primarily due to the thickening of the arytenoid mucosa, with partial or total loss of thyroid cartilage. Reduced posterior airway space can be observed in patients with total or partial laryngectomy, and the association with OSA is explained by the alteration of the normal tongue dynamics in patients undergoing resections with reconstruction [13]. It is well-known that vocal cord paralysis is one of the causes of OSA, and it is widely accepted that any narrowing of the airway can lead to the development of OSA [1,2,14]. Therefore, head and neck cancer is a risk factor for OSA [14].

The coincidence between sleep disorders and head and neck cancer has been studied by comparing tumors with the characteristics of the applied treatment, showing that 50% of patients treated with radiation therapy for oropharyngeal and laryngeal cancer have OSA [15]. Many research studies have shown causality between diagnosis and therapeutic options [14,15]. Head and neck cancers can lead to OSA, and this interdependence, which represents a treatment condition, has not been fully elucidated in many studies.

When it comes to the relationship between cancer and fatigue or insomnia, there are few studies that reflect the relationship with OSA [16]. Symptoms such as daytime fatigue are frequently present, as well as daytime sleepiness and snoring. The patients in the evaluated literature have an active form of cancer, with squamous cell carcinoma and male predominance [16]. The evaluation of clinical supports associated sleep disorders with head and neck cancer and considers tracheostomy as the main therapeutical approach. Clinical oncological results, including mortality, are associated with the values of the AHI, and quality of life is assessed through the analysis of multiple questionnaires, showing the exacerbation of several parameters after therapy [17].

An analysis presenting the physio-pathological mechanisms as well as therapeutic results demonstrates that OSA is more common in patients with tumors in the head and neck area, and this indicates that intermittent hypoxemia contributes to the deterioration of the condition after radiation therapy or chemotherapy. OSA is considered more than just a comorbidity in these patients [15,16,17,18].

### 3.4. Concepts Related to Therapeutic Management in Obstructive Sleep Apnea Syndrome in Patients with Head and Neck Tumors

The evaluation of studies related to head and neck cancer associated with OSA and its appropriate treatment indicates that OSA contributes to tumor growth and metastasis. Numerous mechanisms link OSA to an increased incidence of cancer, and factors such as prognosis, systemic inflammation, sympathetic activity, angiogenesis, and immune alteration can lead to increased mortality and cancer risk [19,20]. There is a high prevalence of sleep disorders in patients with head and neck cancer. Management protocols for OSA in cancer patients are based on the use of laser devices for excising excess mucosa and relieving upper-airway obstructions. Additionally, surgical techniques for glottic stenosis following partial laryngectomy are employed. OSA caused by the narrowing of the retro-lingual space is treated using collation and tunneling at the base of the tongue [20,21].

A study that evaluated comparative data between total and partial laryngectomy with OSA demonstrates a higher incidence of OSA in patients with partial laryngectomy when compared to data on the general population within the same age group [22]. OSA was much more severe in patients with vertical partial laryngectomy compared to those with horizontal laryngectomy [22]. Spirometry methods continue to play an important role in screening patients with partial laryngectomy who have suspected OSA. Some studies reveal specific characteristics of OSA in patients with total laryngectomy conducted for laryngeal cancer, such as a higher rate, the absence of a correlation between OSA severity, age, and body mass index (BMI), and increased use of the Epworth sleepiness scale (ESS) to assess daytime sleepiness. As a result, the quality of sleep and quality of life in patients with partial laryngectomy are influenced without being correlated with daytime sleepiness and obesity [23]. When evaluating patients who underwent supracricoid partial laryngectomy, which aims to preserve laryngeal function in laryngeal cancer, a lower quality of life is observed among patients. Patients with sleep disorders show compensatory mucosal proliferation in the arytenoid region, along with laryngomalacia. The thyroid cartilage, which supports the larynx, is replaced with soft tissue in this section, which has a high collapsing capacity during sleep. Partial laryngectomy disrupts the support and architecture of the hypopharynx and larynx. The soft tissue in the lower part of the larynx has a greater tendency to collapse during sleep [22,23,24].

Radiotherapy and chemotherapy for laryngeal and pharyngeal carcinoma are considered to reduce the need for surgical procedures. However, the discussion also involves the potential side effects, such as dry throat, mucositis, burns, and swallowing difficulties. There is an increased prevalence of OSA before and after radiotherapy. Interestingly, patients with OSA who have undergone radiotherapy and chemotherapy do not necessarily present obesity as a significant factor. Nevertheless, their MRI evaluations show a significant narrowing of the upper airway [25].

The risk of sleep apnea after treatment for head and neck tumors in patients with obesity is produced by the association between hyperlipidemia and specific factors for OSA [26]. The study’s conclusion emphasizes the need for correlating OSA symptoms in cases of cervical tumors. The use of the STOP-Bang questionnaire assessed the risk of OSA, showing an elevated risk in tongue cancer patients, in patients who have positive test results for papillomavirus, and in those who have undergone chemotherapy [26]. The study demonstrates the impact on the majority of patients with throat tumors and an increased prevalence of OSA, but without a clear correlation between tumor location, tumor stage, cancer treatment, and OSA severity [26]. Nevertheless, it is recommended to screen all patients with oropharyngeal tumors for the presence of OSA before and after their treatment [27].

As part of the management of patients with OSA, continuous positive airway pressure (CPAP) therapy is considered. However, CPAP therapy may not always yield the expected therapeutic results and can sometimes worsen upper airway obstruction due to its effects on the epiglottis, leading to epiglottic collapse. Epiglottic collapse caused by CPAP indicates reduced tolerability of this treatment. Therefore, alternative treatment options for patients with tumors may include upper-airway surgery or tracheostomy [28]. Multiple studies indicate a decline in the quality of life of patients with head and neck tumors, especially in terms of voice quality both pre- and post-therapy over a 24-month period. Most studies emphasize the need for comprehensive support, including nutrition, vocal rehabilitation, and swallowing rehabilitation over a 24-month period post-radiotherapy [29]. OSA in patients with tumors who have undergone radiation therapy remains a topic of discussion due to its negative impact on the quality of life [30]. Polysomnography, as a diagnostic method for OSA, shows a lack of correlation between the test results and daytime sleepiness assessed using the Epworth sleepiness scale in the analyzed studies [30].

As for patients with head and neck cancer, voice quality remains inferior post-radiotherapy, but there are studies that demonstrate no statistically significant changes over 24 months [29]. Quality of life mostly remains at the same levels, but the patients most frequently have a dry mouth and sticky saliva. 

## 4. Conclusions

The analysis of multiple studies regarding the correlations between OSA, comorbidities, and head and neck tumors indicates a significantly increased risk of OSA in association with conditions such as diabetes, metabolic syndrome, cardiovascular diseases, and head and neck tumors, particularly laryngeal tumors. This association has a physio-pathological basis. The various surgical methods followed by radiation and chemotherapy for tumor treatment do not exclude the increased risk of developing OSA after treatment. This significantly influences the quality of life of patients who survive these types of tumors.

It is important to continue studies to clarify the influence of laryngeal structures in the development of obstructive sleep apnea and what the ENT surgeon should do. 

## 5. Future Directions

Due to the multiple comorbidities associated with OSA, the extension of polysomnography associated with investigations during sleep, such as drug-induced sleep endoscopy, represents a tendency for early diagnosis of this pathology, and this affects the quality of life in these patients. 

Patients with head and neck cancer are at high risk of developing obstructive sleep apnea, and this is why it is necessary to expand the polysomnographic investigation of these patients after surgical procedures or after radiotherapy and chemotherapy.

Because this study is not a meta-analysis, it has some limitations in terms of the review methods. There are acknowledged gaps in the literature, such as the correlation between laryngeal cancer and other comorbidities associated with obstructive sleep apnea, that could not be addressed due to a lack of data. We have only presented what we consider important to support the main idea. 

In the future, it will be important to provide more studies that evaluate the role of ENT specialists in the early diagnosis of this condition, which is proven to be directly involved with laryngeal cancer and significantly affects quality of life.

## Data Availability

Data sharing is not applicable to this study.

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
