# Peer review of "Comorbidities and Laryngeal Cancer in Patients with Obstructive Sleep Apnea: A Review"

_medicina, 2023, doi:10.3390/medicina59111959_

Round 1

Reviewer 1 Report

Comments and Suggestions for Authors

improve the paper will all the suggestions:

Introduction

  • Provide more background on the prevalence and health burden of obstructive sleep apnea (OSA) globally and any relevant statistics on incidence in the population of interest. Cite epidemiological studies as doi: 10.1007/s00405-022-07466-9.

  • Elaborate on the pathogenesis of OSA - discuss the anatomical and physiological factors that contribute to airway collapse during sleep.

  • Explain the importance of assessing comorbid conditions in OSA patients and how certain comorbidities may be bidirectional risk factors for OSA. Cite studies.

  • Discuss laryngeal cancer specifically - incidence, mortality, standard treatments, and impact on quality of life. Cite epidemiological and clinical data.

  • Expand on the rationale for examining associations between OSA and laryngeal cancer. What evidence is there that laryngeal cancer patients are at higher risk for OSA? Cite relevant studies.

  • Elaborate on the anatomical and physiological effects of laryngeal cancer and its treatment (surgery, radiation, chemotherapy) that can promote airway collapse and OSA.

  • Discuss the potential impacts of OSA on outcomes in laryngeal cancer patients - recurrence, survival, side effects of treatment etc. Cite studies analyzing this.

  • Explain the need for greater focus on diagnosis and management of OSA as a comorbidity in laryngeal cancer patients. What are the current knowledge gaps?

  • State the focus/objective of your review clearly at the end of the introduction.

Methods section:

Literature Search Strategy:

  • List the databases searched (e.g. PubMed, Scopus, Web of Science) and any date or language restrictions.

  • Specify the search terms used, including any Boolean operators or Medical Subject Heading (MeSH) terms. Provide the full search strings.

  • Describe any supplemental hand searching of reference lists or grey literature.

  • Mention the most recent date the searches were conducted and any search updates.

Study Selection:

  • Outline the inclusion and exclusion criteria used to screen studies for relevance, including parameters like study design, sample size, interventions, outcome measures etc.

  • State the numbers of studies screened, included and excluded at each stage, and reasons for exclusion. A flow diagram can help summarize this.

  • Indicate the review process - number of reviewers, blinding, how discrepancies between reviewers were resolved.

Data Extraction and Synthesis:

  • List the specific data points or variables extracted from the studies - patient demographics, OSA/cancer details, comorbidities, treatment data, outcomes etc.

  • Describe any quality assessment of the included studies - risk of bias, study limitations etc.

  • Explain the approach used to synthesize the data, e.g. narrative synthesis, meta-analysis etc. Specify any statistical methods.

  • If meta-analysis was not performed, provide the rationale.

Results section:

  • Provide more quantitative details from the analyzed studies:

    • Number and types of studies included (randomized trials, cohort studies etc.)
    • Study sample sizes, demographic characteristics
    • Prevalence rates of OSA in laryngeal cancer populations
    • Severity of OSA diagnosed (mild, moderate, severe based on AHI)
    • Outcomes analyzed - survival, recurrence, side effects, QOL, etc.
  • Summarize the key findings from studies on OSA and various comorbidities reviewed. Include prevalence rates.

  • Report on any sub-group differences found - e.g. OSA rates by cancer stage, treatment type.

  • Describe the main anatomical factors implicated in OSA among laryngeal cancer patients.

  • Summarize the effects reported of cancer treatments - surgery, radiation, chemotherapy - on prevalence and severity of OSA.

  • Report on any associations found between OSA and outcomes like survival, tumor recurrence, treatment side effects, QOL, etc. in laryngeal cancer patients.

  • Mention any gaps identified in research on OSA as a comorbidity in laryngeal cancer patients.

  • Use quantitative data to support summaries of study results whenever possible.                                                                                                         

  • Discussion section:

    Interpretation of Findings

  • Provide more context when interpreting the main findings on OSA prevalence and severity in laryngeal cancer patients. Discuss the role of quality of life reducted and cite  doi:10.1016/j.chest.2017.01.020 and doi:10.1016/j.jvoice.2021.09.040.

  • Compare the prevalence rates found in this review to studies in general OSA populations or other cancer groups to put the results in perspective.

  • Discuss possible reasons for any subgroup differences in OSA rates noted based on cancer stage, treatment type etc. Speculate but make sure to denote speculation.

  • Analyze how the anatomical changes identified may contribute to airway collapse and OSA in this population.

  • Explain the clinical significance of the findings on OSA and outcomes like survival, recurrence, side effects etc. in laryngeal cancer patients.

Limitations

  • Consider limitations beyond just the studies reviewed - limitations of the review methods, potential biases, lack of meta-analysis etc.

  • Note any issues with heterogeneity between studies that may affect synthesis of the data and conclusions.

  • Acknowledge gaps in the literature that could not be addressed due to lack of data.

Conclusions

  • Restate the main implications of the findings in context of improving patient care and outcomes.

  • Provide specific directions for future research based directly on gaps identified in this analysis.

Comments on the Quality of English Language

none

Author Response

I would like to thank you for the effort for evaluating the manuscript , and to inform you , that we have made the the changes indicated by you, as follows.

In Introduction, we have included the prevalence of Obstructive Sleep Apnea, globally and some relevant statistic. We have evaluate the pathogenesis of OSA, we have discuss about anatomy and physiology.  We have explain the importance of assessing comorbid conditions in OSA patients

About laryngeal cancer , we have specified incidence, mortality, and treatment, and quality of life.  We hae elaborate the anatomical effects of laryngeal cancer on the upper airway, as well as we have introduce information’s about the impacts of OSA on outcomes in laryngeal cancer patients. We have explain how OSA is associated with laryngeal cancer and, are the knowledge gaps.

In the Chapter  of this paper we completed the information with database and we have introduced more keywords, and information about the references list.  We have introduced the important criteria that we have used for this study.

We have completed patients demographic information, and we have explained  how we have made the litures analysis. Because it is a review we have explain the important factors which are associated with OSA, and we have analized the literaturein this context.

In the results section, we have introduced details from the studies. , and we have explained the correlation between OSA and comorbidities, but especially   between OSA and head and neck cancer. We have evaluate the quality of life in patients with OSA and laryngeal cancer, before and after laryngectomy, and we have introduced paragraph about the prevalenceof OSA in patients with laryngeal cancer. We have reported the important outcomes found between OSA and the survival at the patients with laryngeal cancer.

We have introduced discussions , about quality of life in patients with OSA and laryngeal cancer, and we have analyzed how the anatomical changes identified contribute to the upper airwaycollapsin OSA patients. 

At the end of the review , we have introduced information’s about the limits of this review, we have the provide the important directions for the future. 

We would like to thank you,  for your effort to review our paper, and thank you for your support provided through the evaluation, to obtain an article that can be published in this very important Journal.

Thank You very much

Assoc Prof ADRIANA NEAGOS MD PHD

Reviewer 2 Report

Comments and Suggestions for Authors

The paper is a review of OSA and related comorbidities and cancers. The manuscript is rather disorganized thoughout, and improvement can be achieved by adding summarized table or graphics.

- English proof reading is needed throughout the manuscript.

- It is hard to understand why it is necessary to expand the polysomnographic investigation. Please state the reasons.

- More comprehensive explanation of drug induced sleep endoscopy is needed for the readers to understand well, such as how it is performed, what can be seen, and clinical significances, etc.

- In “Obstructive sleep apnea in patients with head and neck cancer” section, how can you evaluate head and neck cancers in establishing correct treatment of OSA?

Comments on the Quality of English Language

English proof reading is needed throughout the manuscript.

Author Response

I would like to thank you for the effort for evaluating the manuscript , and to inform you , that we have made the the changes indicated by you, as follows.

We have organized better the important part from this article . We have introduce the paragraph abot the role of polysomnography for the patients investigations, and we have mentioned the role of this investigation for the OSA diagnosis.

Regarding the correlation between OSA and head and neck cancer , we have introduced more information’s , about the laryngeal role, and the influence of laryngeal tumor in the development of OSA at the patients with tumor.

We would like to thank you,  for your effort to review our paper, and thank you for your support provided through the evaluation, to obtain an article that can be published in this very important Journal.

Thank You very much

Assoc Prof ADRIANA NEAGOS MD PHD

Round 2

Reviewer 2 Report

Comments and Suggestions for Authors

Thank you for the revision.